# Semantics-Aware Cookie Purpose Compliance

## Abstract

In response to stringent data protection regulations, websites typically display a cookie banner to inform users about the usage and purposes of cookies, seeking their explicit consent before installing any cookies into their browsers. However, a systematic approach for reliably assessing compliance between the website-declared purpose and the semantic-intended purpose of cookies (denoted as *potential cookie purpose violation*) has been notably absent. Websites may still, whether intentionally or unintentionally (e.g., due to third-party libraries imported), mis-declare cookies that may be abused for tracking purposes.

We address this gap with COOVER (cookie value examiner). We advocate that the value of the cookie is a more reliable indicator of its semantic-intended purpose compared to other features, such as expires and meta-information, which can be easily obfuscated. COOVER decomposes the cookie value into primitive *segments* representing minimal semantic units, and fine-tunes a GPT-3.5 model to automatically interpret their value-inferred semantics. Based on the interpretation, it classifies cookies into four GDPR-defined purposes. We benchmark COOVER against two widely-used content management providers (CMPs) i.e., CookiePedia and Cookie Script, and the state-of-the-art cookie classifier named CookieBlock. It achieves an F1 score of 95%, significantly outperforming other methods. To understand the *status quo* of potential cookie purpose violation on the web, we employ COOVER to analyze Alexa Top 1k websites. Remarkably, out of 15,339 cookies across these websites, only 3.1% quality as *truly* necessary cookies, while 44.1% of websites suffer from issues of potential purpose violation. Our work serves as a wake-up call to web service providers and encourages further regulatory interventions to rectify non-compliance issues within the web infrastructure.

## 1 Introduction

Browser cookies have become the *de facto* standard in the web ecosystem for maintaining session information for the stateless HTTP protocol. Such information includes the user's login details, personal preferences, customized settings and browsing history. While originally designed to improve functionality and personalize user experiences, cookies have also sparked significant privacy concerns regarding online tracking, as highlighted by numerous previous studies [32, 33, 45, 48–50]. Indeed, as highlighted by Hu et al. [41], only a minimal subset (13.05%) of cookies prove essential for the normal functioning (e.g., session IDs) or security (e.g., OAuth tokens) of a website, whereas the majority are used for tracking users' online activities, e.g., users' browsing history.

Given their privacy sensitivity, most data protection regulations specifically target cookie usage. These include the California Consumer Privacy Act (CCPA) [1], the Digital Personal Data Protection Act (DPDPA) [13], ePrivacy Directive (ePD) [14], and the General Data Protection Regulation (GDPR) [16]. Generally, they mandate that web service providers (i.e., the *data controllers*) disclose their cookie policies to users (i.e., *data owners*), explicitly stating the purposes for which cookies are installed. Both ePD and DPDPA require general cookie consent, with ePD specifically focussing on cookies necessary for the essential functions of websites. GDPR imposes more stringent requirements on obtaining user consent for all cookies. Non-compliance with these regulations can result in significant penalties. For instance, GDPR enforces fines amounting to "*4% of a company's global turnover or €20 million*" [25].

In response to these regulations, various efforts have been dedicated to ensuring alignment between the actual purpose and declared purpose of cookies, which we refer to as *cookie purpose compliance*. Many consent management providers (CMPs), e.g., Cookiebot [15] and Osano [18], automatically generate cookie policies and user consent notices for websites. They analyze website code or cookies to identify cookie purpose based on a maintained database of known cookies and tracking technologies. Since the enforcement of GDPR in 2018, the prevalence of CMPs has steadily risen [40], with 40.4% of websites employing CMPs among Alexa UK Top 500 websites in 2022 [54]. However, the effectiveness of their analysis techniques remains insufficient in terms of precision. A study [47] reveals that only 11.8% of websites relying on CMPs satisfy GDPR requirements. Additionally, users have raised usability and readability complaints on the generated cookie declarations [29, 44].

Another line of research [28, 30, 41, 45] employs machine learning techniques to identify cookie purposes. They primarily rely on primitive attributes (e.g., *HttpOnly*) or meta-information (e.g., name and cookie length) as classification features. However, these features can be ambiguous or prone to obfuscation. For example, Calzavara et al. [30] build a corpus of commonly-used names for authentication-related cookies, and find that 49.5% cookies that share a name in the corpus are used for unrelated purposes. The research problem of *reliably inferring the value-inferred cookie purposes and assessing the cookie usage* remains largely open.

**Our work**. We bridge this gap with COOVER (cookie value examiner), leveraging the insight that the value of the cookie is a more reliable indicator of semantic-intended purpose compared to primitive attributes and meta-information. It analyzes the cookie value, based on which, it can precisely label the cookie for purpose compliance checking. It uses the four GDPR-defined purposes [11] as the labels, namely *strictly necessary*, *preference*, *statistics* and *marketing*, owing to their prevalence in existing cookie banners. With the learned value purpose, these labels can be expanded to encompass additional categories that may emerge as cookie policies evolve or new privacy concerns regarding cookies appear. The challenges COOVER addresses in interpreting cookie value are at least threefold.

*Challenge #1. Lack of unified patterns*. Developers tend to define patterns for their cookies. For example, statistics cookies from Google Analytics start with "*GA1*" or "*GA*", while those from GitHub Analytics start with "*GH*". Even representing similar types of IDs, cookies may vary in length. This diversity impedes the analyzers merely reliant on cookie patterns.

*Challenge #2. Obscure values*. Most cookie values are encoded, hashed or even encrypted for security or integrity reason.

*Challenge #3. Context sensitivity*. Cookies with identical values or patterns may serve distinct purposes across websites. For instance, a cookie value comprised of '*cid*' along with a random number may serve for the statistics purpose when concatenated with a timestamp, but may serve as a necessary cookie when it is with a session state. A *context-aware* analysis by considering the combination of multiple values is desirable.

Coover tackles these challenges with a divide-and-conquer strategy. It breaks down cookie values into primitive units, denoted as *segments*. These segments include plaintexts that can be interpreted using natural language processing (NLP) techniques, or non-plaintexts that require statistical analysis or machine learning to obtain their (partial) semantics (**Challenges #1** and **#2**). Once the segments within a cookie value are obtained, Coover infers the purpose of the cookie with the combined segments (**Challenge #3**). Conceptually, these segments constituting a cookie value are akin to words constituting a sentence. We thus resort to large language models (LLMs), given their power of assimilating syntax and semantics in language-related tasks. We fine-tune an advanced named GPT-3.5-turbo [19] to automatically classify the cookie values.

We evaluate Coover with 2,300 manually labeled cookie values gathered from Alexa Top 3k websites, against two widely-used CMPs, i.e., Cookiepedia [3] and Cookie Script [10], and the state-of-the-art cookie classifier CookieBlock [28]. Coover demonstrates a superior performance, achieving an F1 score of 0.95, significantly surpassing the CMPs and CookieBlock by around 10% and 18%. We also apply it to analyze all Alexa Top 1k websites, to understand the *status quo* of potential cookie purpose violation on the web. It finds that 44.1% of the websites potential suffer from purpose non-compliance issues. Overall, 91.6% of Top 1k websites exist potential violation on cookie management, revealing a concerning situation.

**Contributions**. Our main contributions are summarized as follows.

- **A systematic approach to analyzing cookie purposes.** We propose an automatic approach to assessing value-inferred cookie purpose by interpreting the segments of cookie values. Our work marks the first to incorporate the value analysis for in-depth understanding of the intentions behind cookie usage, and represents a step forward in the assessment of cookie value-inferred purpose.

- **A cookie segments dataset.** We construct a segments corpus of 118,363 segments, extracted and labeled from 51,144 cookie values from Alexa Top 3k websites. It can serve as a foundation for future research in cookie value analysis.

- **Revealing the *status quo* of cookie value-inferred purpose in the web.** We present the landscape of potential cookie purpose compliance of popular websites. Our findings reveal that the current cookie management remains problematic, raising an alert for online users and web service providers.

**Availability**. Our artifacts and datasets are available at https://github.com/CookieValueAnalysis/cookieslearning2024.

## 2  Background and Approach Overview

**Cookie Purposes**. GDPR [11] regulates cookies into four purposes: *Necessary*, *Preference*, *Statistics*, and *Marketing*. Necessary cookies are essential for the basic functionality of a website, like enabling logins or payment processing. Preference cookies enhance user experience by storing choices such as language or volume settings, but are not essential for core website functions. Statistics cookies track user behaviors, such as page visits and error occurrences, to help improve website performance. Marketing cookies are used to deliver personalized advertisements by tracking users' browsing habits across websites. Detailed cookie purpose definition is presented in Appendix C.

### 2.1  Three Phases of Coover

Coover is designed as a three-phase approach, including *cookie value segmentation*, *value-inferred classification* and *potential compliance checking*, as shown in Fig. 1a.

**Phase 1: Cookie Value Segmentation**. This phase aims to extract all potential segments appeared in the cookie values to construct our segments corpus for the later value-inferred classification. It iteratively breaks down the value into plaintexts and non-plaintexts (including *recognizable patterns*, *frequent occurrences*, and *website conventions*), and then use NLTK techniques and the n-gram model to process them respectively. This phase is detailed in Section 3.

**Phase 2: Value-inferred Classification**. This phase automatically analyzes individual segments combinations thereof within a cookie value to infer its purpose. To accomplish this, we fine-tune a large language model (LLM) called GPT-3.5-turbo [19]. This phase is detailed in Section 4.1.

**Phase 3: Potential Compliance Checking**. This phase checks the value-inferred purposes predicted by Coover against the purposes declared by the website provider, so as to detect potential purpose violation. During this process, Coover has to interpret the declared cookie purposes. It uses GPT-3.5 to accomplish this task. This phase is detailed in Section 4.2.

### 2.2  Data Collection

**Cookie Values Collection**. As there lacks a large-scale dataset of cookie values available for us to use, we proceed to create one by ourselves. We build a crawler based on Selenium [22] to retrieve cookies from each website in the Alexa [4] Top 3k list. The crawler navigates through the websites, simulating random clicks on the links within the webpages. Considering that some funtion-related cookies are only installed when the corresponding functions are invoked, e.g., Youtube [26] setting the "*LOGIN_INFO*" cookie upon users' successful login, we invoke the basic functions for the Top 1k websites during the crawling. These functions are detailed in Table 2 in Appendix A.

We use two browser extensions, namely Consent-O-Matic [46] and Cookie-Editor [9], to record all encountered cookies. Consent-O-Matic can automatically accept all cookie purposes usage in a website. Cookie-Editor provides the option to export all collected cookies into the JSON format, including their name, domain, value and other meta-information. In total, we collect 51,144 cookie values among Alexa Top 3k websites as our cookie value dataset.

**Cookie Declaration Collection**. During the crawling process, we also retrieve each website's cookie declarations, including cookie policy or pop-up cookie consent (i.e., cookie banner). For the cookie policy, the crawler uses the Google search engine to pinpoint pages

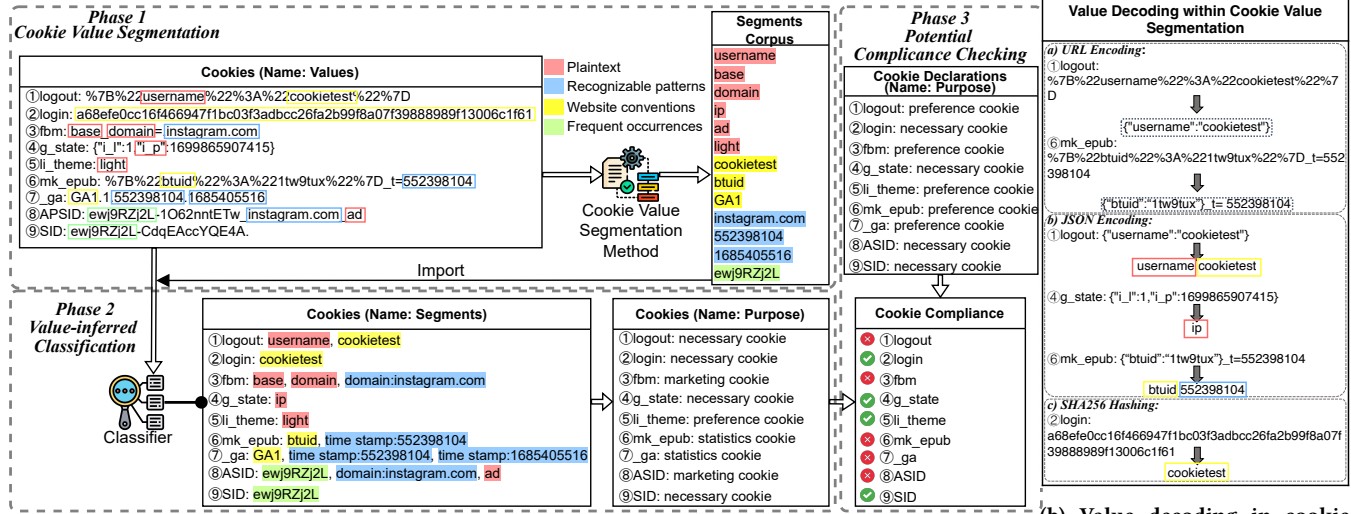

(a) Coover's three-phase approach explained with a running example

(b) Value decoding in cookie value segmentation method

**Figure 1: Overview of Coover**

containing "*cookie*", "*policy*", and "*purpose*" within the website do-main. It sifts through the search results as per Google's ranking, and takes the first page that contains all those keywords as the policy document. If no such page is found after five results, it con-siders that the website has no cookie policy provided. For the cookie consent, the crawler automatically detects the presence of pop-up elements within the retrieved HTML. It extracts the contents pop-up windows that include the keywords of "*cookie*" and "*consent*" as the contents of the website's cookie consent.

## 3 Cookie Value Segmentation

In this section, we summarize the characteristics demonstrated by cookie segments (Section 3.1) and present Coover's segment extraction method (Section 3.2).

### 3.1 Value Segment Characteristics

The cookie value typically comprises several segments, each of which can either be plaintext or non-plaintext. As shown in Fig. 1, the segment "*light*" serves as an example of plaintext, while "*ewj9RZj2L*" exemplifies non-plaintext. Although non-plaintext segments often appear random, they may still contain semantic information. After examining approximately 1,000 segments, we have identified four types into which segments can be classified.

- **Plaintext**. This type of segments exhibit clear semantics. For example, in the cookie "*fbm*" shown in Fig. 1a, the segments "*base*", "*domain*" and "*instagram.com*" directly indicate the browsing history of the user.
- **Frequent occurrences**. Certain strings appear as prefixes or suffixes alongside a random bitstring or ciphertext, poten-tially revealing the purpose of the entire value. For instance, consider Google's usage of the cookies "*SID*" and "*__Secure-1PSID*" as user identifiers. Both cookie values share a com-mon prefix "*ewj9RZj2L*". Recognizing such strings can aid in discerning the semantics of cookies that contain it as a prefix.

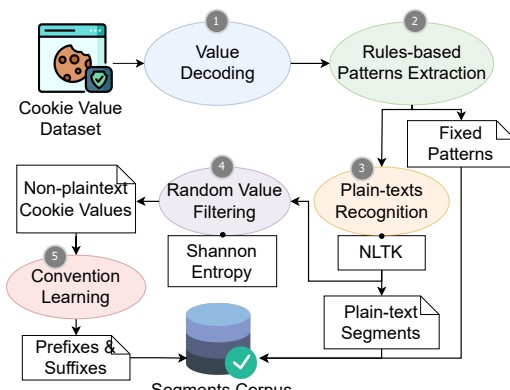

**Figure 2: Coover's cookie value segmentation method**

- **Recognizable patterns**. Some segments follow common structured data formats, such as the GMT format and Unix timestamps. For example, the "*_ga*" cookie employs "1702883 237" to store the user's login time (meaning December 18, 2023, at 07:07:17 UTC). Additionally, certain segments are encoded or hashed using publicly known encoding and hashing algorithms, which can be decoded or cracked. For example, the value "*9f2608067816e38c85edfb0c3985feff32def8b 5dc17bb522ffc2e877e9b386b*" is the SHA256 hash value of "*los angeles*".
- **Website conventions**. Cookies from the same third-party service typically begin with the same semantic strings. For example, the cookie values from Google Analytics invari-ably start with "*GA*", while those from GitHub Analytics start with "*GH*" and end with a timestamp. Certain strings can indicate the source of the cookies, thereby conveying the purpose of the cookie.

### 3.2 Segmentation Method

Considering these characteristics, we devise a segmentation method to identify segments within cookie values. As shown in Fig. 2,

Coover uses five sequential steps in the segmentation process. In this section, we detail each step.

❶ **Value Decoding**. Cookie values are typically processed through encoding, hashing or encryption using publicly-known algorithms. Common encoding methods include JSON Web Token (JWT) format, Base64, and Hexadecimal encoding, all of which can be easily decoded. In contrast, certain values, such as session IDs, are encrypted using robust algorithms like RSA, rendering their ciphertexts infeasible to crack. Some values are hash values generated by publicly-known hash algorithms, which could be cracked with dictionary attacks if the plaintext happens to match an entry in the dictionary. Therefore, in this step, Coover enumerates those commonly used decoding methods for encoded values, including ASCII equivalent decoding, JWT format decoding, URL decoding, Base64 decoding, JSON decoding and Hexadecimal decoding. For hash values, Coover employs an open-source password recovery tool Hashcat [17], which offers a comprehensive range of brute-force and dictionary attacks for deciphering weakly encrypted or hashed strings.

❷ **Rules-based Patterns Extraction**. This step identifies recognizable patterns. Our investigation in Section 3.1 has consolidated four categories of patterns: *a) Date patterns*. Websites use timestamp formats to store the users' browsing dates or the cookie expiration dates, such as GMT format, IANA timezone, ISO 8601 time, and Unix format. We define the regular expressions for each date pattern in Python, e.g., "r"GMT[+−]\d{2,4}" " for GMT date patterns. *b) IP address patterns*. Some websites store users' IP addresses, typically utilising Internet Protocol version 4 patterns. Therefore, we define a corresponding expression as the rule of IP address patterns. *c) UUID patterns*. UUIDs are 128-bit numbers used to uniquely identify a user. The standard, as defined by the Internet Engineering Task Force (IETF), ensures UUIDs are unique across both space and time, even across different devices. They are typically represented as 32 hexadecimal digits, displayed in five groups separated by hyphens, in the form of 8-4-4-4-12. We define a regular expression for the rule of UUID patterns. *d) URL patterns*. Some cookie values store domain addresses to track users' browsing behaviors. For example, the cookie "*fbm*" shown in Fig. 1a stores the base domain "*instagram.com*". Such cookies are mostly plaintexts after the decoding and cracking phase. Therefore, we set corresponding expressions for URL patterns ending with "*com*", "*org*", "*edu*", "*net*", etc. Overall, we define 18 rules for patterns extraction phase (refer to Table 3 in Appendix B).

After the previous two steps, Coover splits each cookie value using commonly-used delimiters, e.g., "|", "&", "=". Each substring is then processed through decoding and cracking. Subsequently, Coover re-applies the two steps until no cookie value can be split, decoded or cracked.

❸ **Plain-text Recognition**. This steps obtains plaintexts appearing in the cookie values, e.g., "username", "base", "light" shown in Fig. 1a. We use the Natural Language Toolkit (NLTK) [21] for this task, leveraging its diverse domain-specific large natural language corpora. Specifically, we rely on its webtext corpus, which contains text data from a variety of web sources. In this step, we manage to recognize 5,302 plaintexts as segments.

❹ **Random Value Filtering**. After recognizing the plain-text segments, there are still 61,618 non-plaintext strings. Among them,

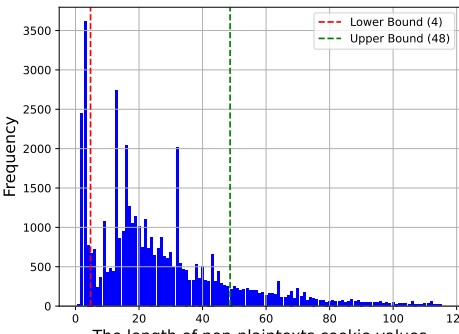

**Figure 3: The length distribution of all split non-plaintext cookie values**

some values exhibit high levels of randomness such that no meaningful segments can be extracted from them. To address this, we employ Shannon Entropy [53] to measure the randomness of a value. Essentially, it measures the uncertainty or unpredictability of potential outcomes from a random variable. High entropy indicates great randomness and low predictability. A previous study [56] investigating the entropy of strings containing repeating patterns reveals that their entropy falls within 75% of the theoretical maximum entropy for strings of equivalent length. Building upon this finding, Coover uses this value as the threshold. When the value's entropy is higher than the threshold, it is filtered out. In this way, 15,241 values are excluded.

❺ **Convention Learning**. After the previous four steps, 46,377 values remain for Coover to recognize. This step aims to learn their (partial) semantics by leveraging website conventions and frequent occurrences. Various algorithms, e.g., the trie tree and suffix tree [38] as well as Recurrent Neural Network (RNN) [58], are capable of learning common texts or strings from a dataset, with many of them using frequency as the learning condition. However, due to the inherent variability of non-plaintext cookie values, some values may appear only a few times, compromising the accuracy of frequency-based detection.

To tackle this, Coover uses an n-gram probabilistic language model [43]. It is well suited to our task, as it can predict the next item in a sequence (with "n" indicating the length of the sequence). We observe that strings corresponding to website conventions and frequent occurrences mostly appear as prefixes and suffixes. For example, in our running example in Fig. 1a, both the cookies "*ASID*" and "*SID*" start with "*ewj9RZj2L*", and all cookie values for Google Analytics start with "*GA*". Thus, we use the length of potential prefixes and suffixes as the learning condition. We analyze the length distribution of 46,377 non-plaintexts values (shown in Fig. 3). Most of them have lengths ranging between 4 and 48 characters. Based on this, we set 25 as the maximum length of potential prefixes and suffixes.

*Redundancy elimination*. The n-gram model may lead to duplicates among prefixes and suffixes. For example, besides identifying the prefix "*ewj9RZj2L*", it may also report "*ew*", "*ewj*", and "*ewj9*". Coover use a redundancy elimination algorithm to handle this, as outlined in Algorithm 1. When the model outputs learned strings from a given value, Coover checks the similarity of each pair consisting of string A of length N and string B of length N-1. The similarity is measured with the Damerau-Levenshtein distance [31],

**Algorithm 1:** Redundancy Elimination Algorithm

---

**Input:** non-plaintext cookie values $V$, the maximum length of learned strings
$\quad$ $LEN\_THRESHOLD$
**include:** JELLYFISH.DAMERAU_LEVENSHTEIN_DISTANCE as DL_DIS
**Output:** $learned\_strings$

1 **Function** Extract_pre_and_suf($V$):
2 $\quad$ $learned\_strings \leftarrow \emptyset, last\_learned \leftarrow \emptyset$
3 $\quad$ **for** $n$ in $(2, LEN\_THRESHOLD)$ **do**
4 $\quad\quad$ $current\_learned \leftarrow$ N-GRAM_MODEL($V, n$)
5 $\quad\quad$ **if** $last\_learned$ **then**
6 $\quad\quad\quad$ $need\_removed \leftarrow$
$\quad\quad\quad\quad$ CHECK_NOISE($last\_learned, current\_learned$)
7 $\quad\quad\quad$ $last\_learned \leftarrow last\_learned.strings \setminus need\_removed$
8 $\quad\quad\quad$ $learned\_strings \leftarrow learned\_strings \cap last\_learned$
9 $\quad\quad\quad$ **if** $n=len\_threshold$ **then**
10 $\quad\quad\quad\quad$ $learned\_strings \leftarrow$
$\quad\quad\quad\quad\quad$ $learned\_strings \cap current\_learned.strings$

11 $\quad$ **return** $learned\_strings$

12 **Function** check_noise($last\_learned, current\_learned$):
13 $\quad$ $need\_removed \leftarrow \emptyset$
14 $\quad$ **for** ($string1, frequency1$) in $last\_learned.items()$ **do**
15 $\quad\quad$ **for** ($string2, frequency2$) in $current\_learned.items()$ **do**
16 $\quad\quad\quad$ $sim \leftarrow$ DL_DIS($string1, string2$)
17 $\quad\quad\quad$ **if** $sim = 1$ & $frequency1 = frequency2$ **then**
18 $\quad\quad\quad\quad$ $need\_removed \leftarrow need\_removed \cap string1$

19 $\quad$ **return** $need\_removed$

---

an extension of the Levenshtein distance [57], which calculates the minimum number of single-character edits required to change one string into the other. For example, the similarity between "*ewj*" and "*ewj9*" is 1. If the distance between A and B is 1, B is considered redundant. To mitigate false positives, COOVER uses their frequency to further confirm whether B is indeed useless. Specifically, if the frequency of A is equal to the frequency of B, B is treated as the redundancy of A. In contrast, if the frequency of B is higher than the frequency of A, B is retained, as it may hold significance in other values. Overall, in step ❺, COOVER obtains 104,719 prefixes and suffixes.

After all these five steps, COOVER keeps all recognizable patterns (step ❷), plain texts (step ❸) and those prefixes/suffixes (step ❹). In total, we obtain 118,363 segments, which consists in our segments corpus. We investigate the capacity of these segments in conveying semantic information detailed in Appendix 5.

## 4 Value-inferred Classification and Potential Compliance Checking

With the segments identified, COOVER proceeds with cookie purpose inference (Section 4.1) and potential compliance checking (Section 4.2).

### 4.1 LLM-based Purpose Inference

The idea of COOVER's purpose inference is to treat segments as the primitive elements of cookie values, and utilize their combinations to infer the purpose of the values. This relationship between segments and cookie values resembles that between words and sentences, where the "meaning" of a sentence stems from the combined semantics of its words. For example, consider the cookies "*SID*" and "*ASID*" in Fig. 1a. Both of them contain the segment "ewj9RZj2L". However, "*SID*" is a necessary cookie, whereas "*ASID*",

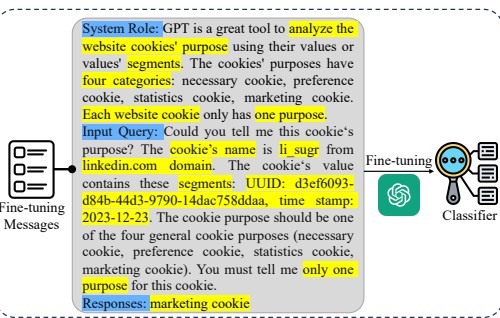

**Figure 4: An example of the fine-tuning message, including the system role, a sample query and the expected response**

which also contains the segments "*instagram.com*" and "*ad*" is used for marketing purposes.

**Model Selection**. COOVER leverages LLMs to capture associations among segments, benefiting from the extensive pre-training that publicly available LLMs undergo. This pre-training enables effective contextual understanding, eliminating the need for complex feature extraction. In this work, we use the GPT-3.5-turbo model due to its accessibility and readiness for fine-tuning.

To evaluate GPT's applicability, we manually assess its performance in purpose recognition. We randomly select 10 short values ($\leq$ 15 characters) and 10 long values ($\geq$ 30 characters), using GPT to identify their purposes. GPT demonstrated proficiency in identifying partial purposes when processing shorter character inputs or longer values with explicit delimiters. However, its performance declined when faced with complex, lengthy sequences, underscoring the need for further optimization through fine-tuning and refined prompt design.

**Model Finetuning**. For fine-tuning GPT-3.5-turbo, we utilize three components: a designated system role, sample input queries, and corresponding expected responses. The system role is specifically tailored for analyzing value-inferred purposes using segments. Additionally, the model is introduced to the four GDPR-defined cookie purposes (Section 2) under the constraint that each cookie serves one purpose. An example fine-tuning message is shown in Fig. 4. The construction of the sample input queries and responses relies on labeled training data. A pilot study (see Appendix D) illustrates that CMPs cannot provide a comprehensive cookie analysis, which is unreliable as training data. Therefore, we use manual labeling for training data. we randomly select 2,300 cookies, and three independent authors manually assign labels to each. To evaluate inter-rater agreement, we apply Fleiss' Kappa [35], achieving a high consistency with a score of 0.978, indicating strong reliability. Subsequently, 88% of the labeled cookies are used as training data for the fine-tuning procedure, and the remaining 12% are reserved for later benchmarking COOVER against baseline analyzers (Section 5).

COOVER uses OpenAI's API for fine-tuning, starting by feeding the fine-tuning message via the `client.files.create()` API, which generates a `training_messages_id` for subsequent use. The fine-tuning process is initiated through the `client.fine_tuning.jobs.create()` API, with `training_messages_id` provided as the `training_file` parameter. This generates a `fine_tuning_job_id`, which is used to track the fine-tuning progress until completion.

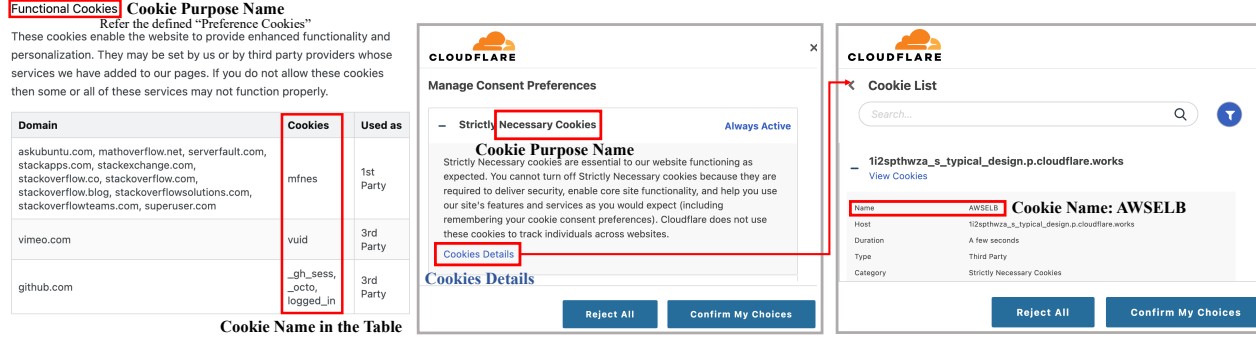

(a) Website cookie policy                                              (b) Website cookie consent notice

**Figure 5: The example of cookie declaration**

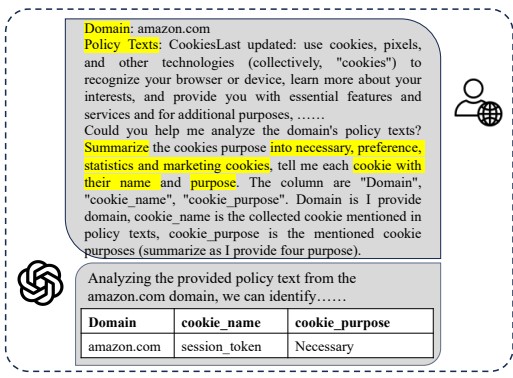

**Figure 6: The prompt example of extracting cookies and their purposes using GPT-3.5**

Coover sets the process to run for 15 epochs. Upon completion, a `model_id` is generated, enabling Coover to access and utilize the fine-tuned model to infer cookie purposes.

### 4.2 Potential Compliance Checking

Most cookie policies outline the collected cookies alongside their names, typically presented in tables, lists or sentences detailing their purpose (an example shown in Fig. 5a). For cookie consent, the cookies' names along with their purposes are presented in the consent notice as shown in Fig. 5b. LLMs are proficient in handling such formatted contexts. Therefore, Coover uses GPT-3.5 again to automatically extract both cookies and their declared purposes within cookie documents.

Coover uses the `BeautifulSoup.html.parser` to parse each policy document, subsequently dividing the text into sentences, with each segment capped at 1000 characters to match to GPT's token limit. As illustrated in Fig. 6, Coover structures prompt by explaining the four GDPR-defined cookie purposes outlined in Section2. The output format is strictly regulated, with placeholders for *domain*, *cookie name*, and *purpose*. Once the prompt is constructed, Coover utilizes the OpenAI `client.chat.completions.create()` API, passing the specified prompt via the `message` parameter. This process enables the extraction of cookies and their declared purposes from each policy statement.

We manually validate the results of cookie declaration extraction to confirm the quality. We randomly select 30 websites from

**Table 1: Performance of Coover vs. Baselines**

|  | Coover | | Cookiepedia | | Cookie Script | | CookieBlock | |
|---|---|---|---|---|---|---|---|---|
|  | Precision | Recall | Precision | Recall | Precision | Recall | Precision | Recall |
| Necessary | 0.86 | 0.90 | 0.83 | 0.90 | 0.83 | 0.71 | 0.31 | 0.86 |
| Preference | 0.93 | 0.96 | 0.93 | 0.48 | 0.78 | 0.78 | 0.52 | 0.85 |
| Statistics | 0.96 | 0.97 | 0.96 | 0.82 | 0.81 | 0.83 | 0.95 | 0.79 |
| Marketing | 0.97 | 0.95 | 0.82 | 0.93 | 0.96 | 0.76 | 0.94 | 0.65 |
| | F1 Score: 0.95 | | F1 Score: 0.85 | | F1 Score: 0.82 | | F1 Score: 0.77 | |

Alexa Top 1k list. For cookies claimed by each website, we randomly choose an equal number of cookies for four purposes. Only 1/240 cookies are mislabeled because of the multiple claims by a website. In particular, the cookie "*VISITOR_INFO1_LIVE*" is claimed as both a preference and statistics cookie by Google but Coover only identifies it as a preference cookie.

### 5 Evaluation

**Baselines and Settings**. We compare the performance of Coover with three baselines, including Cookiepedia [3] and Cookie Script [10], and the state-of-the-art cookie classifier CookieBlock [28]. Cookiepedia and Cookie Script are selected CMPs as they identify a large number of cookies, and high proportion of cookie purposes (see Appendix D). They both annotate cookie purposes based on the domains and names. CookieBlock uses a machine-learning classifier with the input of 21 features extracted in cookie attributes. To apply CookieBlock on our dataset, we re-structure the input cookie data format into the attributes it needs. We use the manually-labeled dataset in Section 4.1 as the benchmark dataset.

**Performance**. Our experimental results are shown in Table 1. Coover achieves significant improvement on all metrics among all cookie purposes. In identifying necessary cookies, Coover can achieve comparable performance with Cookiepedia. They perform substantially better than both Cookie Script and CookieBlock in both precision and recall. It means that for later potential compliance checking, Coover can accurately detect the necessary cookies. For preference cookies, while Cookiepedia obtains similar precision, Coover still has the best result (93% precision and 96% recall), and it drops to 52% precision and 85% recall when using CookieBlock.

CookieBlock remains the high recall on both necessary and preference cookies, however, drops on the precision. We analyze those results and find that all cookies for single sign-on (SSO) login are misclassified as preference cookies rather than necessary cookies.

This problem is also confirmed by [45]. We then confirm the unclassified cookies in both Cookiepedia and Cookie Script, where they have 8 and 21 unclassified labels separately. After removing the cookies labeled as unclassified by two CMPs, we re-calculate the results. We observe that there is no significant change in the result. For example, the F1 score for Cookiepedia and Cookie Script improves from 0.85 and 0.82 to 0.87 and 0.86 respectively. The results still remain worse than Coover.

Coover can identify statistics and marketing cookies with both highest precision and recall. Both statistics and marketing cookies could be used to collect and analyze the users' information and online behaviors. For regulation compliance, such cookies should be clearly declared by websites. CookieBlock can identify these two types of cookies with high precision, however, lower recall. That means it misses some real statistics and marketing cookies in ground truth. We conduct an investigation on this issue, and find it may stem from their unreliable ground truth. For example, it mislabels all "_gat" cookies from Google Analytics [5] as preference cookies rather than statistics cookies. Additionally, even though Cookiepedia have the same precision with Coover on statistics purpose, it also drops on recall. It mis-assigns all cookies of "_ga_{hash_value}" from Google Analytics, e.g., "_ga_KP8QEFW4ML", to the advertising purpose or unclassified purpose, despite their well-known statistics purpose [7].

## 6 Case Study

We apply Coover to study the *status quo* of cookie purpose compliance among the Alexa Top 1k websites. It identifies four types of issues, namely *no cookie usage declaration*, *implicit cookie consent*, *unclaimed cookie usage*, and *potential cookie purpose violation*. In this section, we detail our findings in each of them.

**No cookie collection declaration**. *Around 25.4% of websites fail to provide the cookie declaration for users.* Among Alexa Top 1k websites, 94 websites involve no cookies, and around 50 websites are not English. From remaining 869 websites, we extract 10,846 cookies from our cookie dataset, where 648 websites have cookie policies or cookie consent notices. The remaining 221 websites fail to provide cookie declaration, but they still involve cookies. This violates the regulations like GDPR, which requires website administrators to offer clear and transparent information to users about the use of cookies.

**Implicit cookie consent**. *373 websites only provide cookie policies for users without requesting cookie consents.* We find that 275 websites present cookie consents for users, where some of them are also with the cookie policies. It means that such websites request the users' consent about their cookie usage behaviors. 373 websites presume that users consent to their cookie collection behaviors by default, unless the users proactively access the cookie policy and related settings to make alterations (called *implicit consent*). Such implicit consents violate GDPR, which imposes the cookie regulations about obtaining user consent for cookie usage behaviors, either necessary cookies or not.

**Unclaimed cookie usage**. *511 cookies are unclaimed among 289 websites, where most of them are advertising cookies.* We find that 289/648 websites have some unclaimed cookies, where such cookies are not mentioned in their cookie policies or cookie consented, however, detected and collected by our crawler. Coover

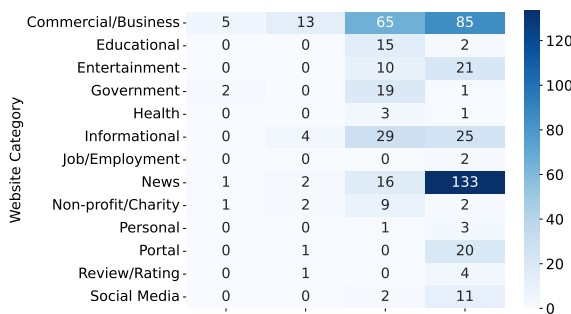

**Figure 7: Unclaimed cookies in each website category**

| Website Category | Necessary | Preference | Statistics | Marketing |
|---|---|---|---|---|
| Commercial/Business | 83 | 68 | 738 | 434 |
| Educational | 20 | 8 | 189 | 20 |
| Entertainment | 10 | 10 | 42 | 146 |
| Government | 14 | 3 | 183 | 1 |
| Health | 14 | 5 | 76 | 50 |
| Informational | 13 | 12 | 195 | 118 |
| Job/Employment | 2 | 0 | 3 | 17 |
| News | 6 | 15 | 202 | 511 |
| Non-profit/Charity | 10 | 7 | 146 | 19 |
| Personal | 2 | 4 | 39 | 25 |
| Portal | 1 | 1 | 18 | 43 |
| Review/Rating | 1 | 3 | 40 | 69 |
| Social Media | 3 | 6 | 11 | 103 |

Detected Purpose Using Coover

**Figure 8: Potential non-compliant cookies in each website category**

identifies these cookies with value-inferred purposes, presented in Fig. 7. Few unclaimed cookies are necessary and preference cookies. Among those cookies, business websites usually fail to declare them. Most unclaimed cookies are statistics and marketing cookies, especially for marketing cookies among news websites. Such cookies violate the GDPR and ePD, which significantly infringe upon user privacy. The marketing cookies are mostly assigned by third-party websites, where websites are obligated to disclose these cookie collection behaviors, even for the first-party websites. Such cookie collection without notification might potentially lead to inadvertent information disclosure by users, such as the leakage of their browsing history and even interests.

**Potential cookie purpose violation**. *3759 cookies are potential non-compliant cookies among 351 websites, where most of them should be statistics cookies.* After identifying the unclaimed cookies, we select the cookies whose claimed purposes are different from Coover-inferred purposes. Among them, we find that a few are claimed for multiple times by the website. For example, Google claims the "NID" cookie as a preference cookie first in their cookie policy, however, re-claims it as a statistics cookie later. In this situation, Coover considers it as the preference cookie for the declaration purpose. Besides these, most potential non-compliant cookies are mistakenly claimed by the websites with one purpose, especially for the statistics cookies.

Fig. 8 displays the overview of potential non-compliant cookies distribution for four cookie purposes in each website category. Similar to the unclaimed cookies, most potential non-compliant cookies should be the statistics and marketing cookies. 3759/10846 observed cookies (34.7%) are non-compliant cookies. For example,

"_ga" cookie for the Google Analytics is commonly used in online websites and widely recognized as a statistics cookie. However, among 648 websites, 179 (27.6%) websites mistakenly claim those "_ga" cookies' purposes (totally 378 cookies), where 177 cookies are claimed as preference cookies and 201 cookies are claimed as marketing cookies.

We identify the distribution of claimed cookie purposes within the value-inferred cookie purposes for these potential non-compliant cookies. 1298 identified marketing cookies using Coover are claimed as the preference cookies in the websites, where most of them are news websites (shown in Fig. 8). For identified statistics cookies using Coover, 907 cookies are claimed as preference cookies, where most of them are business websites. For the claimed necessary cookies, 5 cookies should be the marketing cookies identified by Coover. The results infer that some websites prefer to assign the necessary and preference cookies as statistics and marketing cookies. Necessary cookies can be strictly accepted because the block of them might cause the breakage of websites, indicating that users have to accept those claimed as statistics and marketing cookies. These misguiding declarations might obscure the cookies' behaviors and deceive users, with cookies being professed for alternate purposes. It implies that users might be unable to reject the collection of these cookies, which violates the GDPR.

## 7 Limitations

**Dataset Collection**. Our cookie dataset might be biased for the size of segments corpus of Coover. First, we only collect the general cookies among Alexa Top 3k websites and cookies across functions among Alexa Top 1k websites. This might limit the representativeness of cookie usage patterns for segment corpus. Second, the automatic crawler might differ from the real users' behaviors. The websites can detect the crawler as a bot and provide fake data to the crawler or prohibit it. It will influence the analysis of cookies.

**Cookie Classification**. Without relying on the CMPs, our training and testing dataset for Coover are manually labeled. This may restrict the size of the training set. Additionally, the proportion of necessary and preference cookies is small in our dataset, which causes the unbalanced training set. Especially for the necessary cookies, despite efforts to identify various necessary cookies, their representation remains limited due to the inherent minimal variation in basic website functionality. These may affect the performance of our Coover, compared with the performance of identifying statistics and marketing cookies. Nonetheless, this dataset still provides a valuable basis for training the model.

## 8 Related Work

**Cookie Classification**. Several studies have explored the classification of web cookies using various methods. Calzavara et al. [30] developed a machine-learning model as a classifier to identify the authentication cookies. They used a 2,464 cookies dataset derived from websites among Alexa ranking to analyze authentication cookies. The study confirmed the misclassification of four existing authentication cookie detectors and proposed their classifier based on the cookie value's entropy and length, "httpOnly" attribute, expiration time, etc. Their classifier reached an F1 score of 83% with high recall of 89%.

Hu et al. [41] implemented a cookie name-based classifier using the Multinomial Naive Bayes model, called CookieMonster. It classifies cookies into four purposes, namely necessary, functional, performance and targeting/advertising cookies. CookieMonster is trained with an 11.5k cookies dataset provided by Cookiepedia as the ground truth and achieves more than 94% F1 score. Similarly, Bollinger et al. [28] subsequently applied a machine learning model based on XGBoost to classify cookies collected from websites ranked by Tranco. The classifier used cookie features extracted from cookies' attributes, e.g., the entropy of cookie values, whether cookie value contains URL, 'httpOnly" value, then categorized cookies into those four purposes. Their model, trained on a dataset of 304k cookies from CMPs, achieved a validation accuracy of 84.4%.

Munir et al. [45] focused on first-party tracking cookies identification and proposed a machine-learning based approach. They extracted the structural and flow features from cookies then used filter list and Cookiepedia provided 115k labels to train and test their classifier. The classifier reached an overall accuracy of 90.18% to identify the first-party tracking cookies.

**Cookie Declaration Compliance**. Cookie policies and cookie consents are summarized as the cookie declarations. Many studies have investigated the potential violations of these two documents. Fouad et al. [36] analyzed 20218 third-party cookies with the declared policies, then reported the cookie policies inconsistency among 95% of them. Among around 20k cookies, only 12.85% provided the cookie policies and only 38.38% explicitly presented their descriptions. Santos et al. [51] analyzed the cookie consent among popular English websites in the EU with manual 400 labeled contexts. They then revealed almost 90% of them violated GDPR. Among those cookie consents, 61% had vague purpose descriptions. Matte et al. [44] developed an approach to automatically detect the cookie consent appeared on 560 websites. They indicated that more than half of websites have the violations, where 47% of websites nudged users' consent for all cookies, 7% failed to provide rejections for cookies and 10% used implicit consents for cookie usage. Demir et al. [34] further investigated the cookie consents for "accept and reject" cookies. The results proposed that rejecting cookies might lead to an increase in tracking behaviors from websites.

## 9 Conclusion

In this work, we implement Coover to automatically analyze the cookie values and extracted segments appeared in these values. We build a segment corpus, including around 118k segments. Coover uses the fine-tuned GPT-3.5-turbo model to automatically analyze the semantics for segments in cookie values to predict the value-inferred cookie purposes. The performance of Coover achieves the F1 score of 0.95, higher than the representative CMPs and the state-of-the-art. Coover finally investigates the cookie usage and declaration among Alexa Top 1k websites. The results show that 91.6% Alexa Top 1k websites have potentially violated the GDPR on cookie management.

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

# A  Details on Websites Categories among Alexa Top 1k Websites

In this section, Table 2 introduces the 14 categories for online websites. It details the descriptions of online websites' categories with their basic functions. It is used for the cookie value collection process as mentioned in Section 2.2.

**Table 2: 14 summarized categories for online websites based on SiteSaga [23]**

| Category | Description | Operated Basic Function |
|---|---|---|
| Informational | These websites provide information on various topics. They may be educational, research-oriented, or focused on news and articles. Examples include Wikipedia, online libraries, and news outlets. | search a default topic ("cookie"); browse the first searched link |
| Commercial/ Business | These websites are for businesses and include e-commerce sites where users can purchase products, e.g., eBay and Amazon. | login; search a default product ("Computer"); browse the first searched result; add it into the shopping cart |
| Entertainment | These websites focus on entertainment and include streaming services, gaming websites, celebrity news, and all forms of multimedia content, e.g. Spotify and YouTube. | login; search a default content ("music"); browse the first searched result; add it into the playlist |
| Social Media | Platforms where users can interact, share content, and create a network of contacts. Examples include Facebook, X, and Instagram. | login; browse the first content on the homepage; post a default message ("good") |
| Personal | Blogs, personal portfolios, and personal project websites fall into this category, e.g., Blogger. Individuals often maintain them. | login; click the button element about creating a profile; creating with default contents ("cookietest") |
| Community | Forums and discussion boards where people with similar interests can communicate, ask questions, and share opinions. | login; browse the first content on the homepage; post a default message ("thank you") |
| Government | Websites operated by the government, providing information, services, and resources to the public. | browse the first five contents on the homepage |
| Educational | These websites provide educational content and resources. This includes universities' sites, online courses, and interactive learning platforms. | browse the first five contents on the homepage |
| Health | These websites offer health-related information, telemedicine services, or resources for both medical professionals and the public. | browse the first five contents on the homepage |
| Non-profit/ Charity | Websites of non-profit organizations or charities. These can provide information about their causes and ways to support them. | browse the first five contents on the homepage |
| Portal | These websites offer a variety of services including search engines, email and forums, e.g. Google and Bing. | search a default topic ("cookie"); browse the first searched link |
| News | Dedicated to providing news, these websites can range from general news to niche topics like technology, entertainment, or sports. | browse the first five contents on the homepage |
| Job/Employment | Websites provide opportunities that users can search for job listings, post resumes, or get career advice. | login; search a default content ("tutor"); browse the first searched result |
| Review/Rating | Websites where users can post reviews and ratings of products, services, or businesses, e.g. CNET. | login; browse the first five contents on the homepage |

## B Details on Rules-based Patterns Extraction

Table 3 lists all pre-defined rules for Coover. In phase 1 of Coover, these rules are used to extract the data patterns, IP address patterns, UUID patterns and URL patterns for segments corpus. Section 3.2 details the process for rules-based patterns extraction.

**Table 3: Pre-defined Regular Expressions for Rules-Based Patterns Extraction**

| Categories | Expressions |
|---|---|
| Data Patterns | r"(20(0[0-9]\|1[0-9]\|2[0-9]\|30)-0[123456789]-0[123456789])" |
| | r"(20(0[0-9]\|1[0-9]\|2[0-9]\|30)-0[123456789]-[12]\d{1})" |
| | r"(20(0[0-9]\|1[0-9]\|2[0-9]\|30)-0[123456789]-3[01])" |
| | r"(20(0[0-9]\|1[0-9]\|2[0-9]\|30)-1[012]-[0][123456789])" |
| | r"(20(0[0-9]\|1[0-9]\|2[0-9]\|30)-1[012]-[12]\d{1})" |
| | r"(20(0[0-9]\|1[0-9]\|2[0-9]\|30)-1[012]-3[01])" |
| | r"(20(0[0-9]\|1[0-9]\|2[0-9]\|30)0[123456789]0[123456789])" |
| | r"(20(0[0-9]\|1[0-9]\|2[0-9]\|30)0[123456789][12]\d{1})" |
| | r"(20(0[0-9]\|1[0-9]\|2[0-9]\|30)0[123456789]3[01])" |
| | r"(20(0[0-9]\|1[0-9]\|2[0-9]\|30)1[012][0][123456789])" |
| | r"(20(0[0-9]\|1[0-9]\|2[0-9]\|30)1[012][12]1)" |
| | r"(20(0[0-9]\|1[0-9]\|2[0-9]\|30)1[012]3[01])" |
| | r"(0[123456789]0[123456789]20(0[0-9]\|1[0-9]\|2[0-9]\|30))" |
| | r"(0[123456789][12]\d{1}20(0[0-9]\|1[0-9]\|2[0-9]\|30))" |
| | r"(0[123456789]3[01]20(0[0-9]\|1[0-9]\|2[0-9]\|30))" |
| | r"(1[012][0][123456789]20(0[0-9]\|1[0-9]\|2[0-9]\|30))" |
| | r"(1[012][12]\d{1}20(0[0-9]\|1[0-9]\|2[0-9]\|30))" |
| | r"(1[012]3[01]20(0[0-9]\|1[0-9]\|2[0-9]\|30))" |
| | r"(0[123456789]0[123456789]20(0[0-9]\|1[0-9]\|2[0-9]\|30))" |
| | r"([12]\d{1}0[123456789]20(0[0-9]\|1[0-9]\|2[0-9]\|30))" |
| | r"(3[01]0[123456789]20(0[0-9]\|1[0-9]\|2[0-9]\|30))" |
| | r"([0][123456789]1[012]20(0[0-9]\|1[0-9]\|2[0-9]\|30))" |
| | r"([12]\d{1}1[012]20(0[0-9]\|1[0-9]\|2[0-9]\|30))" |
| | r"(3[01]1[012]20(0[0-9]\|1[0-9]\|2[0-9]\|30))" |
| | r"([01]\d\|2[0-3]):([0-5]\d):([0-5]\d)" |
| | r"(Sun\|Mon\|Tue\|Wed\|Thu\|Fri\|Sat)\+(Jan\|Feb\|Mar\|Apr\|May\|Jun\|Jul\|Aug\|Sep\|Oct\|Nov\|Dec)\+\d{1,2}\+\d{4}" |
| | r"\d{1,2}\s(Jan\|Feb\|Mar\|Apr\|May\|Jun\|Jul\|Aug\|Sep\|Oct\|Nov\|Dec)\s\d{4}" |
| | r"\d{1,2}/\d{1,2}/\d{4}" |
| | r"\d{1,2}:\d{2}:\d{2}\+[APM]{2}", r"GMT[+-]\d{2,4}" |
| | r"\b(?:" + "\|".join(VAILD_TIMEZONES) + r")\b" |
| | r"\b(?:" + "\|".join(CONTINENTS_REGIONS) + r")\/[A-Za-z_]+\b" |
| | r"[+-]\d{2}:\d{2}" |
| | re.compile(r"\b\d{10}\b") |
| IP Address Patterns | r"\b\d{1,3}\.\d{1,3}\.\d{1,3}\.\d{1,3}\b" |
| UUID Patterns | r"[0-9a-fA-F]{8}-[0-9a-fA-F]{4}-[0-9a-fA-F]{4}-[0-9a-fA-F]{4}-[0-9a-fA-F]{12}" |
| URL Patterns | r"https?://(?:[A-Za-z0-9](?:[A-Za-z0-9\-]{0,61}[A-Za-z0-9])?\.)+[A-Za-z]{2,6}(?:[^\s&;]*)" |
| | r"(?:(?:[A-Za-z0-9](?:[A-Za-z0-9\-]{0,61}[A-Za-z0-9])?\.)+(?:com\|org\|net\|edu\|gov\|co\|info\|biz\|io\|app))" |

*VAILD_TIMEZONES = ["EST", "PST", "CST", "MST", "UTC", "EDT", "PDT", "CDT", "MDT"]
*CONTINENTS_REGIONS = ["Africa", "America", "Antarctica", "Asia", "Atlantic", "Australia", "Europe", "Indian", "Pacific"]

## C  Definition of Four Cookie Purposes

Coover uses GDPR-defined cookie purposes [11] as its labels, as they are used as the provenance of reference by most web service providers and CMPs. GDPR places stringent requirements on obtaining user consent for all cookies, including both essential and non-essential cookies. It summarizes four general cookie purposes as follows.

- **Necessary cookies**. These cookies are essential for the fundamental functions of a website, which thus are also called *essential cookies*. They enable core features, e.g., user logins, shopping carts and payment. If such cookies are disabled, the web service may not function normally.

- **Preference cookies**. Preference cookies, which are also called *personalization* or *functionality* cookies by some CMPs, are used to record users' choices to enhance their experiences. Examples of such cookies include language, region, and volume settings for online video players. These cookies are not essential for the website's functionality, but enhance the usability and personalization.

- **Statistics cookies**. Statistics cookies collect information about how online users interact with a website, which are thus called *performance* or *analytics* cookies. Such cookies track user activities like which pages are most frequently visited, how long users stay on a page, and whether users encounter error messages. Website administrators analyze these data to understand how users interact with their websites to enhance users' experience accordingly.

- **Marketing cookies**. Marketing cookies are used to deliver targeted advertisement to users, which are also called *targeting* or *advertising* cookies. They can track users across websites and collect information about their browsing habits. This information is used to create user profiles and show ads that are more relevant to the individual's interests. Such cookies are usually set by advertising networks with the website provider's permission.

## D  A Pilot Study of CMPs

### D.1  Background: CMPs Usage and User Concerns

After the publication of GDPR, cookie declarations are enforced. Thus, more and more websites use CMPs to generate their cookie consents and collect users' agreement for cookie collections. Hils et al. [40] pointed out a rapid increase in CMPs usage after 2018. They found that, among websites in Tranco ranking, moderately popular websites prefer to use CMPs for their cookie collection. A few studies have investigated CMPs and cookie consents from users' perspectives. Toth et al. [55] proposed that CMPs were not able to protect users' privacy even though they claimed GDPR compliance. The purpose of these CMPs for providing the pop-up window with cookie consent is to acquire users' acceptance for websites, not alert users for data protection. Users have raised concerns about cookie declarations. Gray et al. [37] analyzed cookie consents from users' experience and discussed the usage of dark patterns in the contexts of cookie consents. Dark patterns are the design techniques to manipulate users into accepting cookies for website cookie collection.

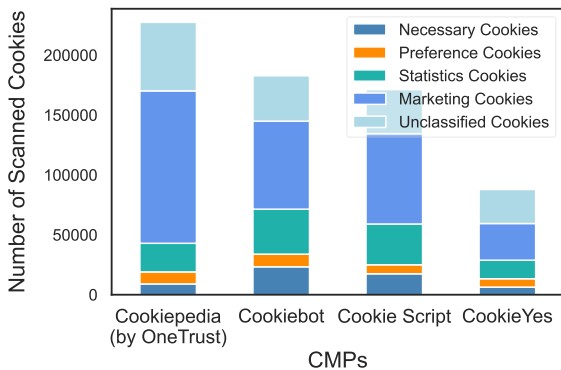

**Figure 9: The distribution of the cookies identified by the four CMPs from Top 10k websites (September to December 2023). The *unclassified* purpose means those cookies whose categories cannot be recognized by the CMP.**

Habib et al. [39] conducted a 1109-participant user study and evaluated several dark patterns used in cookie consents. They indicated the type of dark patterns that could lead participants to accept all cookies and proposed suggestions on these dark patterns.

### D.2  Pilot Study

Since CMPs are widely used by web service providers for cookie scanning and categorization, we conduct a pilot study to explore the effecacy of CMPs in cookie analysis. We select CMPs to study from a pool of options certified by Google [20], considering that developers tend to trust Google-certified solutions. These CMPs come recommended by Google to web service providers who use Google services such as Google Adsense and Ad Manager. Although various CMPs produce cookie banners [40], only a few of them can automatically identify cookie purposes and release their reports for free. From them, we narrow down our selection to those CMPs that have cookie purposes aligning with the GDPR-defined purposes (Appendix C). This process leads us to keep Cookiepedia (provided by OneTrust) [3], Cookiebot [15], CookieYes [12], and Cookie Script [10]. They together account for more than 70% market share, according to a recent report [8].

The four selected CMPs provide the "scanning" function, where we can input the website link and obtain the report of cookies detected by them along with CMP-labeled cookie purposes. To automate this process, we build a crawler using Selenium [22] to interact with each CMP. It feeds the Top 10k websites in Alexa ranking [1] [4] into the CMPs, and obtain the cookie reports. The statistics of the identified cookies is presented in Fig. 9.

**CMPs Efficacy Analysis**. The four CMPs demonstrate considerable *disparity* and *uncertainty* in their reports. First, as shown in Fig. 9, both the number of identified cookies and the distribution of the recognized cookie purposes significantly vary. Taking the marketing purpose for an example, Cookiepedia reports that more than half of all scanned cookies serve marketing purpose. In contract, Cookiebot and Cookie Script show significantly lower proportion, and CookieYes displays an even smaller proportion.

---

[1]Alexa ranking is available at https://github.com/CookieValueAnalysis

Such inconsistencies pose challenges for web service providers in placing trust in the assessment. Second, all CMPs report a large proportion of unclassified cookies, and the proportion ranks as the second highest in almost all of them. This may put their users in the risk of violating GDPR, which mandates the clear declaration of cookie purposes.

**Table 4: A detailed study of four CMPs' reports regarding Google domains**

| CMPs | Scanned Cookies | Necessary Cookies | Preference Cookies | Statistics Cookies | Marketing Cookies | Unclassified Cookies |
|---|---|---|---|---|---|---|
| Cookiepedia | 26 | 0 | 0 | 0 | 13 | 13 |
| Cookiebot | 27 | 4 | 0 | 10 | 13 | 0 |
| CookieYes | 3 | 0 | 0 | 0 | 1 | 2 |
| Cookie Script | 21 | 0 | 1 | 9 | 6 | 5 |

We conduct a manual analysis on the cookies from Google domains, to investigate the *accuracy* of CMPs' labeling. As shown in Table 4, the CMPs identify differing number of cookies, with Cookiepedia identifying 26, Cookiebot 27, CookieYes 3 and Cookie Script 21. Upon close examination of the reported marketing cookies, we find only five of them are consistently identified by Cookiepedia and Cookiebot. We notice two known statistics cookies, i.e., *__utmc* and *_ga* from Google Analytics [5], within this category. Both of them are misclassfied by Cookiebot as marketing cookies, and CookieYes and Cookie Script even fail to identify *__utmc*.

## E  Analysis of Coover-inferred Segments

We explore how the semantics of Coover-inferred segments contribute into the purpose classification by analyzing their embeddings, which serve as crucial inputs for LLMs' classification tasks. As it is infeasible for us to examine each individual segment in our corpus, we adopt a group-based approach. We cluster the embeddings with the hypothesis that *if segments effectively convey semantics, semantically-similar embeddings will cluster together.*

Given that embeddings of segments are high-dimensional, we use the k-means clustering algorithm [42] with the Uniform Manifold Approximation and Projection (UMAP) in our study. K-means automatically cluster embeddings based on their distances, and UMAP can reduce the dimensionality of these embeddings while preserving their intrinsic structure. This enables k-means to identify clusters that might be difficult to discern in the original high-dimensional space [27]. Additionally, UMAP can filter out the noise and irrelevant features of the embeddings, which improves the quality of clusters formed by k-means. In our experiment, the Silhouette score [52] is used to determine the optimal number of clusters that should be generated. Overall, seven clusters are generated, and they are summarized in Table 5. Fig. 10 shows the UMAP visualization of our segments corpus. It confirms the performance of k-means clustering. Fig. 11 visualize clusters for our segments corpus.

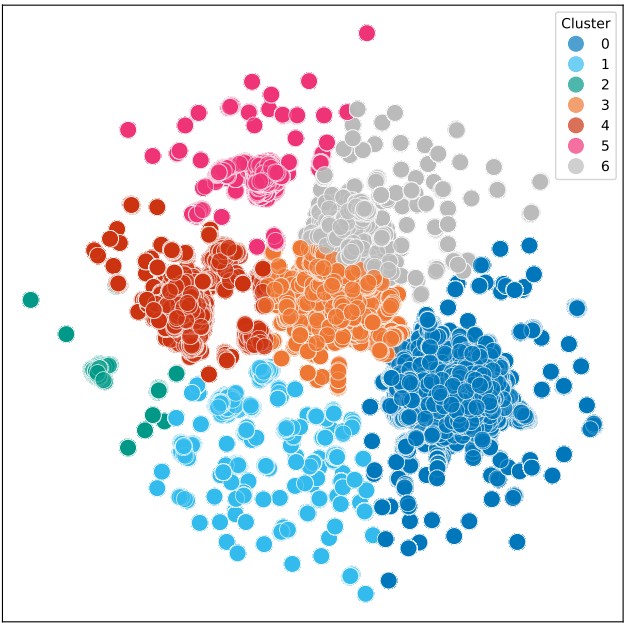

**Figure 10: The UMAP visualization of our segments corpus. The clustering algorithm used is the k-means clustering algorithm.**

**Table 5: The summary clusters in segments corpus**

| # | Potential Semantics |
|---|---|
| Cluster 0 | Most segments are used for statistics purpose, e.g., "sid" for session ID with "sts" for time stamp. |
| Cluster 1 | Most segments serve necessary and preference purpose, e.g., "cookietest6" representing our test account name is stored for login functionality. |
| Cluster 2 | Most segments are UUID that can be used across four purposes, e.g., a UUID stored with a login status can be used for necessary purpose; a UUID set by third-party domain work for marketing purpose. |
| Cluster 3 | Most segments in this cluster contain encrypted strings. Linking to the original cookie values, segments might work for the necessary purpose, e.g., "jot" segment as a part of a cookie value to store session ID securely. |
| Cluster 4 | Most segments contain unique numerical IDs. Third-party domains use such IDs to identify the stored users' information for marketing purpose. |
| Cluster 5 | Most Segments consist of hashed information strings and session number, which cannot be cracked using common methods. They might be used for necessary or statistics purpose, e.g., "0A9997F77942AF1F5412F7148" as a part of "hashed_email" cookie to store the login email. |
| Cluster 6 | Most segments can be linked to a domain to store the users' browsing histories, e.g., "livestream". They are not necessary for a website's basic functions. |

**Cluster 0 (59,935 segments).** Cluster 0 contains segments using object identifier (oid) or session ID (sid) for tracking and analyzing

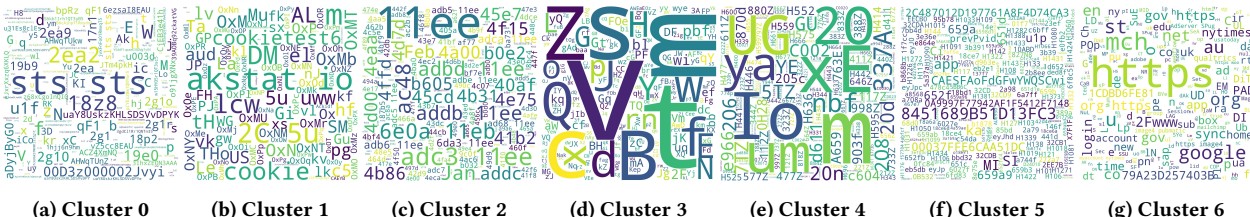

| (a) Cluster 0 | (b) Cluster 1 | (c) Cluster 2 | (d) Cluster 3 | (e) Cluster 4 | (f) Cluster 5 | (g) Cluster 6 |

Figure 11: Word cloud of segments in each clusters

users' browsing history, which is mostly related to statistics purposes. We observe that most cookies that include these segments are session-unique. For example, the segment "*00D3z000002Jvyi*" shown in Fig. 11a is an `oid`, which is a part of "*oinfo*" cookies. When a new session is created, cookies with an `oid` or `sid` are installed to record users' view page, time or status. For example, as shown in Fig. 11a, the segment "*sts*" is the timestamp to represent the time that the session starts or is last accessed. These cookies can store the users' browsing period or even visit frequency (as a counter), which are useful for analyzing a user's interest within a domain.

**Cluster 1 (6,416 segments).** Segments in this cluster are mostly used for recording users' settings, which are related to the necessary or preference purposes. First, some cookies store the user' name for authentication, such as "*cookietest6*" in Fig. 11b, which is the test account names COOVER uses for login. As another example, the "*id_token*" cookie stores the special token of given name of the user. It also contains the issuer of the token (a random string), the intended recipient of the token ("*cms-auth-proxy*" segment in this cluster), and the time of authentication. It is apparently used for authentication purpose. Second, some segments in this cluster are used for recording the users' preference settings. They store the users' region or language settings, e.g., the keyword "*EU*", "*AU*", "*JP*" (shown in Fig. 11b). Such cookies are preference cookies.

**Cluster 2 (3,162 segments).** Most segments (2,754) in this cluster are UUIDs, used as unique session ids or user ids to track sessions. For example, Washington Post [24] uses the "*sec_wapo_login_id*" cookie to identify the login status, storing "67c2c979-5e69-45f1-b1fd-1e4534e46939" segment for a user in this cluster. It is automatically generated and kept after a user logs in.

Third-party domains may also set the UUID to track users across different websites for the marketing purpose (targeted advertising) or the preference purpose (the specific settings in a domain for users). For example, "*11ee*" shown in Fig. 11c is a part of "37e8de40-ad6e-11ee-8579-291dad710f11" as the value for the "*IMRID*" cookie. This cookie is used to measure viewing and clicking of online advertisement for Nielsen [6]. Nielsen advertising automatically generates this cookie for each user. Additionally, some websites use CMPs to collect and save users' settings lead to additional cookies being installed. For example, OneTrust uses the "*OptanonConsent*" cookie to store users' cookie selection. The segment "*Jan*" shown in Fig. 11c is part of the timestamp for such cookies with UUID and other settings, such as version code and landing page.

**Cluster 3 (19,077 segments).** Most segments in this cluster are the complex strings in a cookie value. Such cookies are generally ciphertexts, beyond the capability of the common cracking methods that COOVER uses in its segmentation. After tracking back the original cookie values, we find that these cookies generally work

for session management and security (necessary purpose). Compared with segments for session management in the previous two clusters, segments in these cookie values have no fixed format (like UUID or stored as "sid"). These segments and cookie values might encrypt specific session information (e.g., session ID) in a way that is decryptable only by the server that sets the cookie.

**Cluster 4 (11,920 segments).** There are 6,096 digit strings and 2,013 timestamps in this cluster. The digit strings may stand for user IDs or server IDs. Some websites may use digit strings rather than alphanumeric strings due to configurations in their backend servers. When websites use such values, they usually set a long one to ensure uniqueness across an extremely large number of users or sessions. This is particularly relevant for websites with a vast user base or those handling massive amounts of data, where the probability of duplicate IDs must be minimized. For example, segment "6987981977721894400718623" in this cluster is a part of the "*demdex*" cookie value for Adobe Audience Manager [2].

**Cluster 5 (7,399 segments).** This cluster consists of some hashed personal information and session numbers. For example, "0A9997F77 942AF1F5412F7148" in Fig. 11f, as a part of the "*hashed_email*" cookie, is the hash value of an email address. Moreover, this cluster also contains some segments for session numbers. In contrast with session ID segments in Cluster 3, a random string for session ID, segments in this cluster are generated using a specific formatting rule defined by a website. For example, the "*ioam2018*" cookie which has the segment "000142218ebf296e66027e79a" always has its value starting with "000".

**Cluster 6 (10,454 segments).** This cluster contains segments about web links (shown in Fig. 11g). When the user interacts with a webpage, some cookies store the clicked website links for statistics purposes, e.g., "*https*", "*edu*", "*google*" in Fig. 11g. Websites generally use such cookies to store the users' visited webpages with the generated ID, then analyze the users' online behaviors. They mostly are not necessary for a website's essential function.

