# OpenReview forum: "Semantics-Aware Cookie Purpose Compliance"
_ACM.org/TheWebConf/2025/Conference — WWW 2025 Oral_

### Official Review · Reviewer_UtbS · 2024-11-27

**Novelty:** 7
**Technical Quality:** 6

**Review:**

### Quality:
The paper explores the potential misdeclaration of cookies that may be used for tracking purposes. The research methodologically seems to be robust, employing fine-tuned GPT-3.5 for cookie purpose analysis and achieving strong performance metrics (F1 score of 0.95). Additionally, creating and including a manually curated dataset allows for reliability.

### Clarity:
The paper's structure clearly states the methodology, findings, and objectives. There are detailed sections such as the classification process which give more clarity to the paper.

### Originality:
The research focuses on addressing a gap in cookie compliance evaluation and introduces/ uses fine-tuning a large language model as a new method/way to achieve cookie compliance.

### Significance:
Their findings showcase that there is a widespread non-compliance with cookie regulations (only 475 of the 15339 cookies across Alexa's top 1k websites are necessary) which emphasizes the need to improve regulatory measures.


### Pros:
1. The introduction of COOVER: a new approach for analyzing cookie values for classification using LLMs
2. Manual dataset creation of a large-scale cookie value dataset (51,144 cookies, 118363 segments).
3. Highlighting rampant non-compliance amongst Alexa's Top 1k websites

### Cons:
1. Possible dataset bias
2. Although tested, nowhere does it show how Coover performs in real-world applications where cookies change frequently.
3. The dependence on automated crawlers: Understandable but could possibly limit the ability to capture encrypted cookies

**Questions:**

1. The researchers mention a possible bias in the cookie dataset. How can this bias be mitigated?
2. If Coover were to be tested in real-world applications, how would it cope with the changing cookies?

**Reviewer Confidence:**

4: The reviewer is certain that the evaluation is correct and very familiar with the relevant literature

**Scope:**

4: The work is relevant to the Web and to the track, and is of broad interest to the community

---

### Official Review · Reviewer_McsE · 2024-12-02

**Novelty:** 5
**Technical Quality:** 5

**Review:**

## Summary
The main content of the article is about an automatic approach called Coover to assess the purpose of cookies by analyzing their values. The authors propose that analyzing the value of a cookie is a more reliable indicator of its purpose compared to primitive attributes and meta-information. Coover uses LLMs to extract and classify the purpose of cookies based on their values. The authors evaluate the performance of Coover and find that it achieves a higher F1 score compared to other methods. They also apply Coover to analyze the cookie usage and declaration among Alexa Top 1k websites, finding that a large percentage of these websites potentially violate GDPR regulations on cookie management.

## Strengths
1. Analyzing the purpose of cookies is important for users to understand their function and protect their privacy.
2. The dataset constructed in the paper is valuable for future research.
3. The experimental results validate the effectiveness of the proposed method in the paper.

## Weaknesses
1. The dataset and method code in the paper have not been released, making it difficult to reproduce the work in future studies.

**Questions:**

N/A

**Reviewer Confidence:**

2: The reviewer is willing to defend the evaluation, but it is likely that the reviewer did not understand parts of the paper

**Scope:**

4: The work is relevant to the Web and to the track, and is of broad interest to the community

---

### Official Review · Reviewer_FAKh · 2024-12-02

**Novelty:** 5
**Technical Quality:** 5

**Review:**

The paper presents a novel tool called Coover for automatically extracting and segmenting cookie values and classifying them according to the GDPR-defined purposes, for ensuring cookie compliance and potential purpose violation.

Strengths:
The three-phase approach (cookie value segmentation, classification, and compliance checking) is detailed well, with a running example.
The segmentation approach proposed aims to address existing challenges of cookie pattern-based analysis, length variation, encoding and context-sensitive use, by analysing a large corpus of cookies and identifying segment characteristics.
The paper creates a dataset of 51,144 cookie values and 118,363 labeled segments, which could benefit future research in the domain.
The empirical evaluation demonstrates the system's effectiveness, with Coover achieving an F1 score of 0.95, outperforming existing cookie classifiers like CookieBlock, Cookiepedia, and Cookie Script.

Limitations:
The evaluation relies heavily on manually labeled data, which, while reliable, may not scale effectively to larger datasets without similar manual effort.
Details about potential biases introduced during the fine-tuning process or challenges in managing ambiguous cookies are not deeply discussed.

**Questions:**

1. What are the ethical implications of using brute force attacks for deciphering hashed strings, esp. in the case of user login cookies?
2. The Related Work section mentions other benchmark cookie datasets? Are these open source? Why was Coover not assessed on these datasets? This would have validated its performance on data other than the authors' curated dataset.
3. Can Coover be applied to analyze and validate compliance in other domains of web tracking, like fingerprinting or local storage, which may not rely on cookies?

**Reviewer Confidence:**

3: The reviewer is confident but not certain that the evaluation is correct

**Scope:**

4: The work is relevant to the Web and to the track, and is of broad interest to the community

---

### Official Review · Reviewer_mKmP · 2024-12-03

**Novelty:** 5
**Technical Quality:** 5

**Review:**

This paper proposes a systematic approach COOVER for reliably assessing compliance between the website-declared
purpose and the semantic-intended purpose of cookies (denoted as potential cookie purpose violation).
Coover decomposes the cookie value into primitive segments representing minimal semantic units, and fine-tunes
a GPT-3.5 model to automatically interpret their value-inferred semantics. Based on the interpretation, it classifies cookies into four GDPR-defined purposes.

## Pros

1. Novel Framework: The use of cookie values for compliance checking is a novel contribution.
2. Thorough Evaluation: Achieving an F1 score of 95% demonstrates strong results.
3. Real-World Relevance: Analysis of cookie compliance for Alexa Top 1k websites provides valuable insights for industry and regulators.

## Cons

1. Applicability to smaller or less structured websites is not very clear.
2. Lack of Alternative Approaches: The paper does not compare the LLM-based approach with more explainable or lightweight models.
3. Ethical Concerns: The paper does not deeply address ethical concerns related to automated compliance analysis and potential misuse.

**Questions:**

1. Ethical Implications: Have you considered the ethical implications of automating compliance checks? Could this lead to misuse by less-regulated entities?
2. Robustness Against Obfuscation: How does Coover perform against deliberate obfuscation or encryption of cookie values by malicious actors?
3. Comparison with Lightweight Models: Have you considered comparing Coover with simpler or more interpretable models for cookie classification?

**Reviewer Confidence:**

3: The reviewer is confident but not certain that the evaluation is correct

**Scope:**

3: The work is somewhat relevant to the Web and to the track, and is of narrow interest to a sub-community